# *Toxoplasma gondii* Serotypes in Italian and Foreign Populations: A Cross-Sectional Study Using a Homemade ELISA Test

**DOI:** 10.3390/microorganisms10081577

**Published:** 2022-08-05

**Authors:** Sara Caldrer, Ambra Vola, Guglielmo Ferrari, Tamara Ursini, Cristina Mazzi, Valeria Meroni, Anna Beltrame

**Affiliations:** 1Department of Infectious, Tropical Diseases and Microbiology, IRCCS Sacro Cuore Don Calabria Hospital, Negrar di Valpolicella, 37024 Verona, Italy; 2Microbiology and Virology Unit, IRCCS San Matteo Hospital, 27100 Pavia, Italy; 3Centre for Clinical Research, IRCCS Sacro Cuore Don Calabria Hospital, Negrar di Valpolicella, 37024 Verona, Italy; 4Molecular Medicine Department, University of Pavia (PV), 27100 Pavia, Italy

**Keywords:** *Toxoplasma gondii*, serotype ELISA, Italians, migrants, epidemiology

## Abstract

*Toxoplasma gondii* is a protozoan parasite responsible for human toxoplasmosis. The three major clonal lineages and different recombinant strains of *T. gondii* have a varied global distribution. This study aimed at evaluating the epidemiological distribution of types II and I–III and recombinant or mixed *T. gondii* in Italians and foreigners residing in Italy, establishing an association between serotypes and demographic characteristics. We collected the sera of 188 subjects who had tested positive for specific *T. gondii* antibodies. The population was differentiated into groups based on sex, nationality, and place of birth (Italy, Africa, South America, Asia, or Europe (except Italy)). We then performed a homemade ELISA test that detected both the antibodies against the amino acid sequences of the three main genotype antigens (I–III) in human sera and discerned the *T. gondii* strains. Serotype II of *T. gondii* was the most prevalent in the Italian population, whereas type I–III was the most prevalent in the foreign group. Surprisingly, we observed a notable amount of recombinant or mixed serotypes in European and Italian subjects. Moreover, we showed a significant difference in the prevalence of *T. gondii* serotypes between men and women, Italians, and foreigners. This descriptive study is the first to investigate the epidemiological distribution of *T. gondii* serotypes in humans in Italy using a homemade ELISA. We considered this technique suitable for discriminating between serotypes II and I–III and, consequently, for an epidemiological study focusing on the observation of circulating *T. gondii* strains and clinical correlations.

## 1. Introduction

*Toxoplasma gondii* is a protozoan parasite responsible for human toxoplasmosis (HT) that chronically infects more than one third of the global population [1]. Besides the three major lineages (types I–III), 16 haplogroups of *T. gondii* belonging to six ancestral groups have also been described using whole-genome sequencing analyses [2,3,4]. These sub-clusters may be associated with geographical origins and phenotypic characteristics [5]. Moreover, there are still atypical isolates with unique polymorphisms that cannot be categorized as one of these haplogroups, probably due to recombinant strains [6].

The three major clonal lineages of *T. gondii* and the different recombinant strains have varied global distributions [7]. The types II and III strains predominate in Europe and North America [8], whereas the atypical strains are most prevalent in South America [9]. The maximum level of genetic diversity is found in the wild Amazonian areas, where the ancestors of current *T. gondii* populations emerged 1.5 million years ago [9]. The wide variety of host animals may also explain the high level of diversity and pathogenicity of *T. gondii* in Brazil [10] resulting from recombination events. In Africa [11] and Asia [12], haplogroups have been described in addition to the three most common clonal lineages, named Africa 1, 2, and 3 for the former, and Chinese 1 for the latter.

Illustrating the geographical distribution of *T. gondii* strains worldwide is important given its strong correlation with clinical manifestations [1]. Strain I has demonstrated high virulence, likely due to its high in vitro motility and ability to cross cellular barriers, with rapid in vivo dissemination [13,14]. Types II and III, on the other hand, have shown much lower levels of pathogenicity [15,16]. In South America, more attention has been placed on atypical strains responsible for severe forms of acute *T. gondii* infection and ocular HT in immunocompetent subjects [17,18,19,20,21,22].

Traditional strategies to identify a *T. gondii* strain have typically focused on inter-strain differences at the genomic level [23,24]. PCR analyses of restriction fragment length polymorphisms (PCR-RFLP) or microsatellite markers have permitted an epidemiological description of *T. gondii* strains worldwide [2]. However, this approach requires isolating enough parasitic DNA, which can be difficult to obtain and involves invasive sampling. Such sampling techniques include amniocentesis or the collection of vitreous fluid and often provide an insufficient amount of DNA. However, serological tools using peptides can exploit the strong and persistent humoral immune response to *T. gondii* [25] as the parasite protein antibodies remain throughout the patient’s life. Moreover, evidence has shown the humoral response to *T. gondii* is strain-specific [26]. Antibodies are produced against immunodominant parasite proteins, which polymorph according to the *T. gondii* strain. New serologic tests have been produced to recognize differences in the antibody profiles generated by different *T. gondii* strains [27,28].

In Italy, the seroprevalence of *T. gondii* infections ranges from 12% to 22% in women of childbearing age and from 13.8 to 34.3% in pregnant women [29,30,31]. Although genotyping studies of *T. gondii* have never been conducted on humans, molecular surveys on wildlife have demonstrated the circulation of genotypes I–III and the recombinant or mixed (R/M) genotypes in the Italian animal reservoir [32,33,34,35,36,37,38].

The aim of the present descriptive study was to assess the epidemiological distribution of *T. gondii* serotypes in Italians and foreigners residing in Italy using a homemade enzyme-linked immunosorbent assay (ELISA), as well as to establish the association between serotype and demographic characteristics.

## 2. Materials and Methods

### 2.1. Study Design and Setting

This was a cross-sectional study conducted at the Department of Infectious, Tropical Diseases and Microbiology (DITM), IRCCS Sacro Cuore Don Calabria Hospital, Negrar di Valpolicella, and at the SC Microbiology and Virology Department, IRCCS Policlinico San Matteo, Pavia. These two institutions are primary referral centers for HT in pregnancy and for parasitic diseases in Italy.

### 2.2. Study Population and Data

We analyzed samples from 188 patients with symptomatic HT or from mothers and newborns screened for toxoplasmosis who were referred to our center between 1 January 2008 and 1 September 2020.

As inclusion criteria for the cases, we considered (1) all serum samples stored at −80 °C with (2) *T. gondii* IgG antibody titer ≥ 100 IU/mL and (3) availability of demographic characteristics. The variables considered were sex, age, and nationality. The overall population was differentiated into groups based on the following demographic characteristics:Sex: female or male;Nationality: Italian (patients born in Italy) or foreign (patients born in other countries);Place of birth: Italy, Africa, South America, Asia, or Europe (except Italy).

All subjects provided written informed consent to donate their biological samples for research purposes.

The study protocol was approved by the two competent Ethics Committees (Comitato Etico per la Sperimentazione Clinica delle Provincie di Verona e Rovigo on 11 November 2020—protocol n. 59897; and the Comitato Etico per la Sperimentazione Clinica di Pavia on 22 January 2021—protocol n. 20210006505).

### 2.3. Identification of T. gondii Antibodies in Sera Samples

Different ELISA tests were used in the laboratories of the two centers. Specifically, the VIDAS Toxo IgG II assay (Biomerieux-Mercy l’Etoile—France) and the ISAGA IgM assay (BioMérieux, Craponne, France) were used for *T. gondii* IgG and IgM. The results were confirmed by the LIAISON^®^ XL Toxoplasma IgG and IgM (DiaSorin, Saluggia, Italy) and ARCHITECT Toxo IgG and IgM assays.

Finally, all the sera samples were tested with VIDAS Toxo IgG II, which permitted the same evaluation in IU. Positive samples with a *T. gondii* IgG antibody titer equal to or above 100 IU/mL were considered for serotype determination.

### 2.4. ELISA Protocol for the T. gondii Serotype Determination

The homemade ELISA was developed in the parasitology laboratory of the Fondazione IRCCS Policlinico San Matteo, following the procedure proposed by Kong J.T. et al. [27], with minor modifications. This test could distinguish the three main genotypes (I–III; II; R/M) in all collected human sera.

Lyophilized synthetic peptides were derived by synthesizing the amino acid sequences present in the dense granules and antigens of different *T. gondii* strains. Cysteine and keyhole limphet hemocyanine (KLH) were added to the N or C terminus of each peptide to allow coupling to the carrier and to promote adhesion to the plate to perform the ELISA test (Primm srl, Peptide Synthesis Milan). The 6-I/III-KLH and 6II-KLH peptides were used to identify serotypes I/III and II, respectively. As a positive control, a whole antigen from the sonication of *T. gondii* was used, provided by Diasorin.

Specifically, the peptides coupled to KLH were diluted to 1 µg/µL in PBS. Fifty microliters (0.5 mg) of each peptide or of the total *T. gondii* lysate (0.005 µg/µL) was loaded into each well of a polystyrene ELISA plate (Falcon flat-bottom ELISA plate) for overnight coating at 4 °C. Each well was then blocked with 100 µL of a 3% bovine serum albumin (BSA) PBS with 0.01% thimerosal, and the plates were held for one hour at room temperature. Next, 50 μL of diluted serum was added to each well, and the plates were incubated at room temperature for one hour. The serum dilution was chosen according to the titer of the sample. After washing, 100 μL of goat antihuman IgG (Sigma-Aldrich, St. Louis, MO, USA) at a final concentration of 1/2000 in PBS-BSA 3% was added for 1 h at room temperature. After washing once more, 100 μL of staining solution was added (citric acid, Na_2_HPO_4_, H_2_O_2_, and o-phenylenediamine dihydrochloride) and the plate was incubated at room temperature in the dark for 15 min. The optical density (OD) was red at 450 nm. The results were expressed in OD. The cut-off value was then calculated as the median OD of the negative control plus two standard deviations. A non-reactive (NR) result was registered when the serum sample had an OD index below the cut-off value (obtained by testing 31 samples from 31 seronegative/non-immune patients in duplicate), with a positive response against the toxoplasma antigen [39].

Based on the ELISA test results and depending on the antibody reaction to the different *T. gondii* strains, the serum samples were classified into four serotype subgroups: I–III, II, and R/M, when a positive response was registered against all the peptides; or NR.

### 2.5. Statistical Methods

Statistical analyses were performed using STATA software v14.0 (StataCorp LP, TX, USA) or SAS EG v7.1 (SAS Institute Inc., Cary, NC, USA). Images were generated with GraphPad Prism v8.3.0 (GraphPad Software, San Diego, CA, USA). Continuous variables (age of subjects) were summarized with the median and interquartile ranges (Q1–Q3), while count variables (*T. gondii* serotypes, sex, or place of birth) were summarized with absolute and percentage frequencies. Non-parametric tests were applied according to data distribution. A chi-square test with adjusted *p*-value was used to evaluate the relationship between the observed results with categorical variables (sex or place of birth). The Bonferroni correction was used when performing multiple tests. The Kruskal–Wallis test, a rank-based nonparametric test, was used to determine statistically significant differences between two or more groups for an independent variable (*T. gondii* serotypes) on a continuous dependent variable (age of subjects).

## 3. Results

### 3.1. Demographic Characteristics of the Overall Study Population Included

The analysis was conducted on sera from 188 subjects with a median age of 32 years (Q1 (first quartile)–Q3 (third quartile) 23.5–40), 107 females (56.9%) and 81 males (43.1%). The population was composed of 103 Italians (54.8%) and 85 foreigners (45.2%). The geographic origin of the overall sample was heterogeneous, as shown in Figure 1A,B. Most subjects were born in Italy (103 cases; 54.8% of the population), Africa (50; 26.6%), or South America (20; 10.6%), with a minority born in other regions (10 in Europe and 5 in Asia) (Figure 1B). Ghana was the most reported African country with 11 cases (5.9%), followed by Nigeria with 8 cases (4.3%). For South America, nine subjects were from Brazil (4.8%) and five subjects were from Colombia (2.7%). Five subjects (2.7%) were born in Romania, the largest number of European subjects evaluated (Appendix A).

A statistically significant difference between the median age of the two subgroups was found (35 (22–51) and 31 (24–36.5) years for Italians and foreigners, respectively; *p* = 0.008). No other statistically significant differences were found between the demographic data of the two subgroups. Females accounted for 56.3% of the Italian group and 57.7% of the foreign group. Minors (under 18 years of age) represented 17.5% (18/103) and 11.8% (10/85) of the Italians and foreigners, respectively.

### 3.2. Distribution of T. gondii Serotypes in the Overall Sample and in the Italian, Foreigner, Male, and Female Subgroups

First, we analyzed the prevalence of serotypes among the samples studied (Figure 2A). A total of 55 of the 188 subjects analyzed (29.3%) had serotype I–III, 55 (29.3%) had serotype II, and nine (4.8%) had the R/M. A total of 69 sera (36.7%) resulted in NR (Table 1).

Regarding the male and female prevalence stratified by *T. gondii* serotype, serotype II was observed in 31 of 107 (29.0%) females and in 24 of 81 (29.6%) males, with serotype I–III in 29 (27.1%) females and 26 (32.1%) males (Figure 2B, Table 1). A higher prevalence of the R/M serotype was found in the male group (6.2%) than the female one (3.7%). The NR result was registered in 43 (40.2%) females and 26 (32.1%) males.

A statistically significant difference (*p* < 0.001) was found in Italian and foreign prevalence stratified by *T. gondii* serotype (Figure 2A, Table 1). In fact, 42 of 103 Italians (40.8%) and 13 of 85 foreigners (15.3%) had serotype II, whereas serotype I–III was found in 15 Italians (14.5%) and 40 foreigners (47.1%). The R/M serotype was found in four Italians (3.9%) and five foreigners (5.9%), and the NR result was observed in 42 Italians (40.8%) and 27 males (31.7%).

Subsequently, the prevalence of each *T. gondii* strain was assessed within the nationalities of the patients (Figure 2A) (*p* < 0.001). In Europeans (*n* = 10), strain II was found in three subjects (30%), I–III in two (20%), and the R/M in two (20%), whereas three (30%) had the NR. In contrast, in the Africans (N = 50), strain I–III was identified in 24 subjects (48%), II in 6 (12%), R/M in 2 (4%), and the NR result in 18 (36%) subjects. Among the patients born in South America (N = 20), 13 (65%) had serotypes I–III, 2 (10%) had II, 1 (5%) had the R/M, and four (20%) had the NR. Finally, two patients of Asian origin (N = 5) had serotype II (40%), while one had strain I–III (20%) and two (40%) the NR. No cases of the R/M strain were identified in the Asian group.

Analyzing the women enrolled (N = 107) according to nationality, a significant difference was found in the strain distribution between Italians (*n* = 58) and foreigners (*n* = 49) (*p* < 0.001) (Figure 2B females, Table 1). In fact, *T. gondii* strain II was found in 24 of 58 Italian females (41.4%) compared to 7 of the 49 foreign females (14.3%). Conversely, 24 foreigners had strain I–III compared to 5 Italian females (49 vs. 8.6%). The R/M strain was identified in three foreign females and one Italian female (6.1 vs. 1.7%), while the NR result was observed in 15 foreign and 28 Italian females (30.6 vs. 48.3%).

For the men (N = 81), the distribution of serotypes was not significantly different between the Italians and foreigners (*p* = 0.084). However, strain II was found in 18 of 45 Italians (40.0%) compared to 6 of the 36 foreigners (16.7%) (Figure 2B, males; Table 1). Conversely, 16 foreign males had strain I–III compared to 10 Italians (44.4 vs. 22.2%). The R/M strain was identified in three Italian and two foreign males (6.7 vs. 5.6 %), whereas the NR strain was present in 12 foreign and 14 Italian males (33.3 vs. 31.1%).

### 3.3. Demographic Characteristics According to T. gondii Serotype

The sex, age, nationality, and place of birth of the included patients are shown in Table 2 and in Figure 3 for each *T. gondii* serotype identified.

Subjects with serotype I–III (N = 55) were predominantly female (52.7%) and foreign (72.7%). The majority were Africans (43.6%), Italians (27.3%), and South Americans (23.6%), with only two from Europe (3.6%) and one from Asia (1.8%). Subjects with serotype II (N = 55) were also largely represented by females (56.4%), yet mostly Italian (76.4%). Among the 10 foreigners, 6 were from Africa (10.9%), 3 were from Europe (5.4%), 2 were from South America (3.6%), and 2 were from Asia (3.6%). The small number of subjects with R/M strains (N = 9) were predominantly male (55.6%) and foreign (55.6%). Among the latter, two were from Europe (22.2%), two were from Africa (22.2%), and one was from South America (11.1%). Four of the nine subjects with the R/M serotype were Italian (44.4%). For each serotype, no statistically significant difference was found in the proportions of the two sexes. However, the median age of the subjects who tested positive for the R/M serotype was higher (45 years) compared to those who tested positive for other serotypes, although this difference was not statistically significant (30 and 31 years for I–III and II, respectively, *p* = 0.068).

## 4. Discussion

This descriptive study is the first to investigate the epidemiological distribution of *T. gondii* serotypes in humans in Italy using a homemade ELISA. Most of the subjects were Italians, followed by Africans and Latin Americans. As regards sex, women accounted for 57% of the population. The foreigners were younger than the Italians, reflecting the demographic characteristics of the immigrant population in Italy (https://www.istat.it/it/archivio/224943 and https://www.istat.it/it/archivio/249445; published at 13 December 2018 and 26 October 2020 respectively).

In the overall study population, serotypes II and I–III (29.3% each) were detected most frequently, while the R/M serotype only represented 4.3% of all samples. As expected, serotype II was most prevalent in the Italian population (40.8%), whereas type I–III was most prevalent in the foreign group (47.1%), particularly the South American population (65.0%). These data are similar to those reported in the literature. In fact, in Europe, type II predominance has already been observed [40,41], with type III only occasionally found and type I rarely so [42]. R/M strains are especially rare in Europe and presumably due to contamination with non-European strains from traveling or the consumption of imported food [39,43]. Recently, severe HT in immunocompetent travelers from Africa caused by a strain belonging to the African 1 lineage was reported in France [44]. Surprisingly, 20.0% of the European subjects and 4.0% of the Italians analyzed in our study had a recombinant or mixed serotype, unlike 5.9% of the foreigners and only 5.0% of the South Americans. Studies have shown that these strains are dominant in South America because of genomic recombination that generates diversity [2,39,45,46]. However, while the data obtained from the Italian subjects were valid, the data from the Europeans may have been influenced by the small sample (*n* = 10).

In our African population, the most prevalent serotypes were I–III (48.0%) and II (12.0%), in line with the literature [47], and the remainder were R/M and NR (4.0% and 36.0%, respectively). A recent review by Galal et al. created the first partial mapping of *T. gondii* in Africa [11]. In addition to the type II and III lineages, the coexistence of additional clonal lineages, known as African haplogroups 1 to 3, has been described in western and central Africa [48,49]. As recombinants of highly prevalent lineages, these haplogroups may not have been recognized by our ELISA and were consequently included in the NR group. 

With regards to Asia, only five subjects (two from Bangladesh, one from Nepal, one from Russia, and one from Lebanon) were analyzed in our study, thereby impeding any epidemiological conclusion. The first reports from China, Sri Lanka, and Vietnam have revealed major parasite strain uniformity, with types II or III or the Chinese I strains predominating [12]. Despite the high prevalence of HT in Southeast Asia [50], the strain characterization is incomplete. An increasing number of publications have reported genotypes circulating in diverse provinces of China, but very little data are available for other Asian countries [2,51,52]. Consequently, new epidemiological studies on *T. gondii* serotypes focused on these areas are urgently needed.

The circulation of unknown strains may explain the high percentage (36.7%) of NR found in the present study, which also occurred in a previous study [39]. While this could have been due to laboratory errors or poor sera conservation resulting in repeated thawing, the hypothesis that strains were unrecognized by our ELISA remains plausible given that only two peptides were tested. Surprisingly, women predominantly had the NR serotype (40.2%). Considering the place of birth, a large difference in NR proportions was evident between Italian (48.3%) and foreign women (30.6%). This difference is difficult to explain through technical issues alone. The females may have had a higher proportion of NR because they acquired a *T. gondii* infection through different means than the males and, as a result, had a novel, unknown serotype. Finally, the sensitivity of the test may have influenced these results.

Another notable result of our study was the difference in *T. gondii* serotype prevalence between men and women and between Italians and foreigners. Serotype II prevailed in the female group (29.0%), while serotype I–III prevailed in the male group (32.1%). Furthermore, the percentage of R/M serotypes in the male group was higher than the female (6.2 vs. 3.7%). A large difference was evident when stratifying only the Italian population by sex. In fact, 1.7% of females and 6.7% of males had the R/M serotype. This sex difference was also evident for serotype I–III, as 8.6% of females and 22.2% of males had it, but not for the Italians (41.4 vs. 40.0%, respectively). These results support the previous hypothesis of different infection pathways in Italians of both sexes. Interestingly, this sex difference was not found in the 47.1% of the foreign population who had serotype I–III and in the 5.9% with R/M. These findings suggest that Italian males may have contracted the infection by traveling to a country where the different strain circulates or by ingesting food from these areas that contained the parasite cysts [43,44]. Furthermore, it is also plausible that, because the I–III and R/M strains are more virulent than strain II and cause more severe symptoms, they lead infected individuals to hospitalization, which in turn may have caused bias. Importantly, while the females in this study were mostly pregnant and thus enrolled during the screening for toxoplasmosis or had asymptomatic acute HT during the pregnancy, the males performed the serology test for *T. gondii* due to symptoms, such as a case of fever after returning from tropical areas or chorioretinitis. It is important to bear in mind that our hospitals are reference centers for tropical and ocular diseases.

The results of our study have several limitations. First, the study found a high number of NR results. Since the subjects clearly had *T. gondii* infections, this could be interpreted as a limitation of the homemade ELISA, which may have been unable to identify all the strains. Alternatively, this may have been due to the limited sensitivity of the two-peptide method employed. A recent serotype study used two different peptides for each clonal lineage, demonstrating a strong agreement between the serotype and the genotype for infections due to archetypal strains but suggested the need to find new peptides to distinguish type I, type III, and non-archetypal strains [25]. Second, the ELISA did not allow for differentiation between serotypes I and III and the atypical serotypes, so peptides for more precise characterization should be considered in future studies. Thirdly, molecular confirmation for all the serological findings was lacking. Fourth, there may have been a selection bias for the females, as they were probably asymptomatic while pregnant, and for the males because they were mostly symptomatic. This suspicion cannot be confirmed as we had only clinical judgments and not clinical data. Finally, although the study was conducted in two Italian reference centers for HT in pregnancy and parasitic diseases, it was conducted on a small population, preventing the generalization of the results.

Nonetheless, the peptide serological tool exploits the strong and persistent humoral immune response to *T. gondii* infection and can therefore identify the differences in the *T. gondii* strain without requiring parasitic DNA isolation. This feature is essential for performing large-scale epidemiological studies on *T. gondii* serotypes in poorly described geographical areas and for following the flow of *T. gondii* strains in migrants and travelers.

In conclusion, this is the first study that describes *T. gondii* serotypes in Italians and foreigners from countries where epidemiological knowledge remains incomplete. Multicenter studies with better-characterized peptides using a larger cohort of patients could enhance the sensitivity, specificity, and subsequent use of our homemade ELISA. This would improve the epidemiological understanding of *T. gondii* serotypes. Further studies should also consider investigating the correlation between the strain and clinical data of all enrolled patients.

## Figures and Tables

**Figure 1 microorganisms-10-01577-f001:**
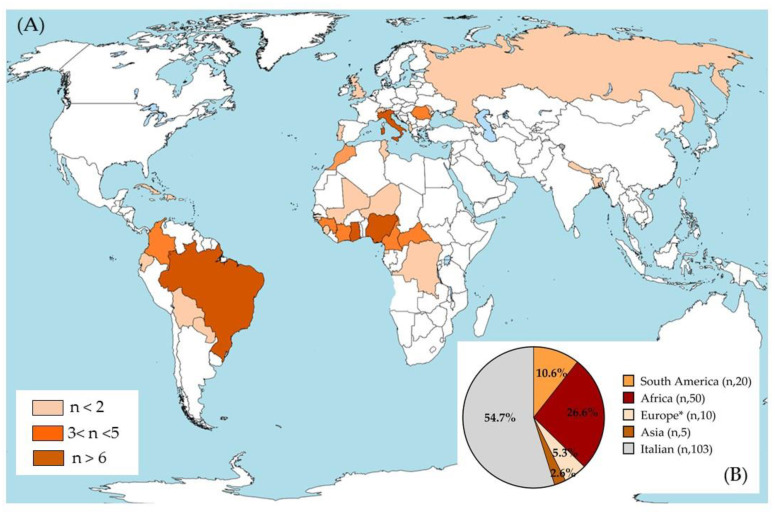
Geographical distribution of the population studied. (**A**) The three colors represent the total number of isolates for each country, as defined in the legend (*n* = sample size). (**B**) The pie chart represents the sample size for each continent and the relative percentage of the population compared to all subjects included in the study. * Europe without Italy. This image was modified starting from the file BlankMap-World-v6.png from the Wikimedia Commons free media repository; GNU Free Documentation License.

**Figure 2 microorganisms-10-01577-f002:**
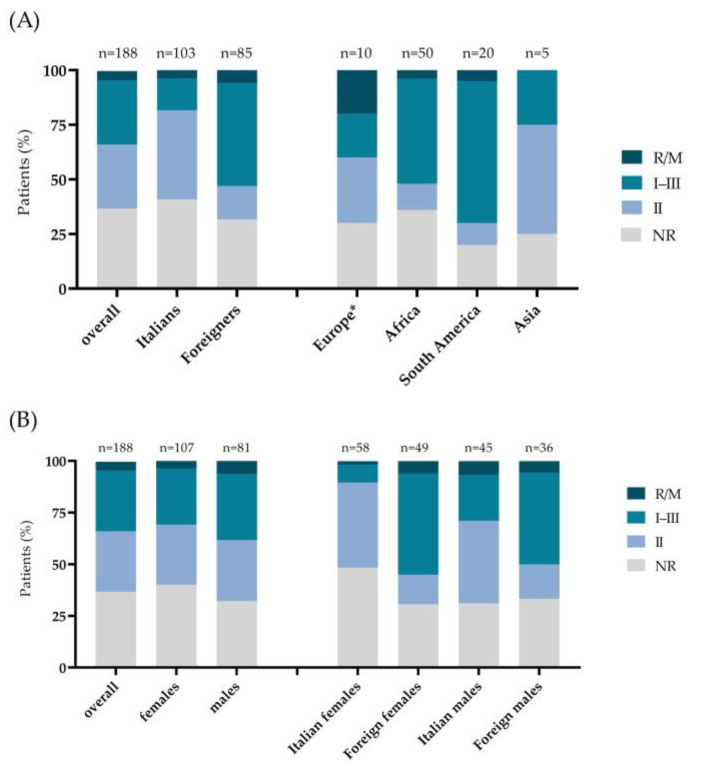
Distribution of *T. gondii* serotypes (**A**) in the overall population stratified by nationality (Italians, foreign) and place of birth (Europe, Africa, South America, Asia) and (**B**) in the overall population stratified by sex (females, males) and by sex and nationality (Italian females, foreign females, Italian males, foreign males). *p*-values are based on the Chi-square test. * Europe without Italy.

**Figure 3 microorganisms-10-01577-f003:**
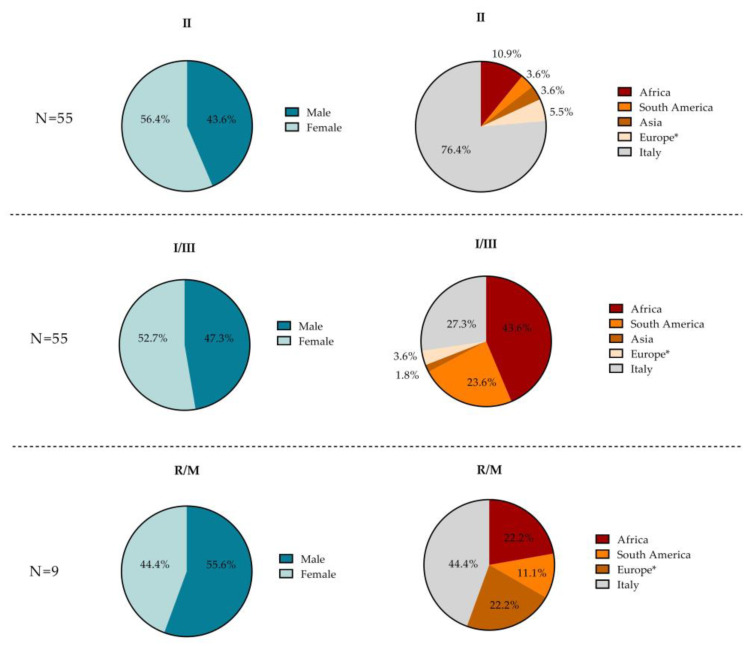
Pie charts showing the patient characteristics (sex and place of birth) for each *T. gondii* serotype (I–III, II, R/M) found in the overall study population. * Europe without Italy.

**Table 1 microorganisms-10-01577-t001:** Prevalence of Italians, foreigners, males, and females, stratified by *T. gondii* serotype.

*T. gondii* Serotypes	Females N = 107 *n*/N (%)	Males N = 81 *n*/N (%)	*p* ^§^	Italians N = 103 *n*/N (%)	Foreigners N = 85 *n*/N (%)	*p* ^§^	Italian Females N = 58 *n*/N (%)	Foreign Females N = 49 *n*/N (%)	*p* ^§^	Italian Males N = 45 *n*/N (%)	Foreign Males N = 36 *n*/N (%)	*p* ^§^
I–III, N = 55	29/107 (27.1)	26/81 (32.1)	*p* = 1.000	15/103 (14.5)	40/85 (47.0)	***p* < 0.001**	5/58 (8.6)	24/49 (49.0)	***p* < 0.001**	10/45 (22.2)	16/36 (44.4)	*p* = 0.218
II, N = 55	31/107 (29.0)	24/81 (29.6)	*p* = 1.000	42/103 (40.8)	13/85 (15.3)	***p* = 0.001**	24/58 (41.4)	7/49 (14.3)	***p* < 0.011**	18/45 (40.0)	6/36 (16.7)	*p* = 0.113
R/M, N = 9	4/107 (3.7)	5/81 (6.2)	*p* = 1.000	4/103 (3.9)	5/85 (5.9)	*p* = 1.000	1/58 (1.7)	3/49 (6.1)	*p* = 1.000	3/45 (6.7)	2/36 (5.6)	*p* = 1.000
NR, N = 69	43/107 (40.2)	26/81 (32.1)	*p* = 1.000	42/104 (40.8)	27/85 (31.8)	*p* = 0.904	28/58 (48.3)	15/49 (30.6)	*p* = 0.306	14/45 (31.1)	12/36 (33.3)	*p* = 1.000

N is the number of non-missing values. ^§^ Chi-square test. Bold *p*-values indicate significant associations after Bonferroni corrections.

**Table 2 microorganisms-10-01577-t002:** Main patient characteristics according to *T. gondii* serotype.

	*T. gondii* Serotypes				
	I–III	II	R/M	NR	*p*
Overall *n*/N (%)	55/188 (29.3)	55/188 (29.2)	9/188 (4.8)	69/188 (36.7)	
Females—*n*/N (%)	29/55 (52.7)	31/55 (56.4)	4/9 (44.4)	43/69 (62.3)	*p* = 0.632 ^§^
Age, years median (Q1–Q3)	30 (23–44)	31 (21–38)	45 (34–57)	33 (27–38)	*p* = 0.068 ^‡^
Foreigners—*n*/N (%)	40/55 (72.7)	13/55 (23.6)	5/9 (55.6)	27/69 (39.1)	*p* < 0.001 ^§^
Place of birth—*n*/N (%)					*p* < 0.001 ^§^
Europe	2/55 (3.6)	3/55 (5.4)	2/9 (22.2)	3/69 (4.4)	
Asia	1/55 (1.8)	2/55 (3.6)	0 (0.0)	2/69 (2.9)	
South America	13/55 (23.6)	2/55 (3.6)	1/9 (11.1)	4/69 (5.8)	
Africa	24/55 (43.6)	6/55 (10.9)	2/9 (22.2)	18/69 (26.1)	
Italy	15/55 (27.3)	42/55 (76.4)	4/9 (44.4)	42/69 (60.9)	

N is the number of non-missing values. ^§^ Chi-square test. ^‡^ Kruskal–Wallis test.

## Data Availability

Not applicable.

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
