# Peer review of "Toxoplasma gondii Serotypes in Italian and Foreign Populations: A Cross-Sectional Study Using a Homemade ELISA Test"

_microorganisms, 2022, doi:10.3390/microorganisms10081577_

Round 1

Reviewer 1 Report

Major comments

The submitted manuscript evaluated the epidemiological distribution of type II, I-III, and recombinant or mixed T. gondii in Italians and foreigners residing in Italy. The authors used a homemade ELISA test to detect antibodies against the three main genotype antigens (I, II-III) in human sera. Overall, the text is easy to follow, clear and indicates the first study to investigate the epidemiological distribution of T. gondii serotypes in humans in Italy.

I believe that the manuscript can still be improved and English should be revised. These changes will make a strong manuscript.

Minor comments

I would like to include some comments and suggestions for the authors, so they can judge or consider to refer to them during the revisions:

Line 55: In-vitro instead in-vitro

Study population and data: Authors could provide the time interval at which patient samples were collected.

Figure 1: If possible, it would be important to increase the resolution of figure 1.

Author Response

We have carefully read all the comments received and would like to thank you for all your suggestions and constructive comments, which allow us to improve our manuscript.

-I believe that the manuscript can still be improved and English should be revised. These changes will make a strong manuscript. 

In agreement with the reviewer’s observation, Brian William Hawkins has performed further English editing.

Minor comments. I would like to include some comments and suggestions for the authors, so they can judge or consider to refer to them during the revisions:

- Line 55: In-vitro instead in-vitro. Done in the text, line 67

- Study population and data: Authors could provide the time interval at which patient samples were collected. Done in the text “who referred to our center from January 1st, 2008, to September 1st, 2020.” Line 105.

- Figure 1: If possible, it would be important to increase the resolution of figure 1. Done. Image replaced with a higher resolution one

Reviewer 2 Report

This manuscript provides the first insight of T. gondii serotypes in humans in Italy. The study’s findings are of importance. Nevertheless, I have the following comments.

Abstract

11.  In lines 30-32, please be more specific regarding serotypes.

Materials and Methods

11. The exact number of the studied population must be added.

22. Please provide some chronological information about the samples used in the study. For instance, they were collected during a …… period.

33. Please explain how the required sample size was estimated.

34. In lines 107-113, the use of different commercial ELISA tests is described. Please provide a more sufficient explanation for the purpose of their usage. 

55.  I suppose that samples characterized as positive by commercial ELISA kits were used in the homemade ELISA test. If so, this should be stated that more clearly in the text.

66. How many peptides were used? In discussion (line 291), the usage of two peptides is mentioned. This information should be added.

67. Which peptides were used?

78. Please explain line 131 “The serum dilution was chosen according to the titer of the sample”.

89.  Please explain why the cut off value was calculated as the median OD of the negative control plus two standard deviations?

. 10. Please add numbers to sub-sections.

Results

11.   In Tables 1 and 2, p values should be estimated for each individual serotype, instead of estimating a p value for all serotypes.

 Discussion

11. In line 327, it is mentioned that the present study was conducted in a small geographical region of Italy. If so, this should be mentioned in Materials and Methods section. Furthermore, the results shouldn’t be generalized by referring to the total Italian population and the text should be modified accordingly (ex. Title, abstract etc.).

22. In line 319-320, please specify the number of NR.

33. In line 320, please explain “chronic T. gondii infections”.

14. In lines 323-324, please add the outcome of the study.

Author Response

We have carefully read all the comments received and would like to thank you for all your suggestions and constructive comments, which allow us to improve our manuscript.

This manuscript provides the first insight of T. gondii serotypes in humans in Italy. The study’s findings are of importance. Nevertheless, I have the following comments.

Abstract

  •  In lines 30-32, please be more specific regarding serotypes.  Done in the text “Discriminating between serotypes II and I-III “ line 31

Materials and Methods

  •  The exact number of the studied population must be added. Done in the text: “We analyzed samples from 188 patients with symptomatic HT”.  line 103
  • Please provide some chronological information about the samples used in the study. For instance, they were collected during a …… period. Done in the text “who referred to our center from January 1st, 2008 to September 1st, 2020. “ lines 104-105
  • Please explain how the required sample size was estimated . We considered all the subjects recruited in the study period, and then we analyzed all the samples considering the inclusion and exclusion criteria. However, the number obtained was higher than that in other studies.
  • In lines 107-113, the use of different commercial ELISA tests is described. Please provide a more sufficient explanation for the purpose of their usage. Different ELISA tests are in use in the laboratories of the two centers. In agreement with the reviewer, we have added: “Finally, all the sera samples were tested with VIDAS Toxo IgG II test, which permitted the same evaluation in IU” lines 148-149
  • I suppose that samples characterized as positive by commercial ELISA kits were used in the homemade ELISA test. If so, this should be stated that more clearly in the text. See line 127: Only samples with sera titers > 100 IU were included in the study
  • How many peptides were used? In discussion (line 291), the usage of two peptides is mentioned. This information should be added. Done in the text “The 6-I/III-KLH and 6II-KLH peptides were used to identify serotypes I/III and II, respectively. As a positive control, a whole antigen from the sonication of T. gondii was used, provided by Diasorin.” lines 160-162
  • Which peptides were used? Done in the text “The 6-I/III-KLH and 6II-KLH peptides were used to identify serotypes I/III and II, respectively. As a positive control, a whole antigen from the sonication of T. gondii was used, provided by Diasorin.” lines 160-162.
  • Please explain line 131 “The serum dilution was chosen according to the titer of the sample”. Sera with Higher IU titer were used at higher dilution to overcome the observable possibility of interference. Data not shown
  • Please explain why the cut off value was calculated as the median OD of the negative control plus two standard deviations? We used the same calculation as other papers, such as Sousa 2017 [39] line 268-269
  • Please add numbers to sub-sections. Done in the text line 95-102-141-150-273

Results

  •  In Tables 1 and 2, p values should be estimated for each individual serotype, instead of estimating a p value for all serotypes. Done in the tab

 Discussion

  •   In line 327, it is mentioned that the present study was conducted in a small geographical region of Italy. If so, this should be mentioned in Materials and Methods section. Furthermore, the results shouldn’t be generalized by referring to the total Italian population and the text should be modified accordingly (ex. Title, abstract etc.   This is a fair observation. We cannot say that our study was conducted on a population that fully represents the Italian population as a whole; however, the two referral centers involved in the study protocol are referral centers for HT in pregnancy and parasitic diseases in Italy. In fact, our centers treat patients from all over Italy. We emphasized in line 100 the referral role of our centers.
  • In line 319-320, please specify the number of NR. NR value is present in the table 1, table 2 and in the text “A total of 69 sera (36.7%) gave a NR result (Tab. 1).” line 175,320,324,362….
  •  In line 320, please explain “chronic T. gondii infections”.In agreement with reviewer ‘s observation we deleted chronic line 483
  •  In lines 323-324, please add the outcome of the study. We have added the outcome: “A recent serotype study used two different peptides for each clonal lineage, demonstrating a strong agreement between the serotype and the genotype for infections due to archetypal strains, but suggested the need to find new peptides to distinguish type I, III, and non-archetypal strains. Lines 515

Reviewer 3 Report

This is an interesting preliminary study on antibodies against specific genotypes of Toxoplasma affecting patients in Italy.

There are some questions that need to answered regarding the methodology;

1. the protocol mentioned newborn being sampled - this would be a very distinct patient group, how were the dealt with in Results (there is no specific mention)

2. The ELISA protocol is missing a detailed description of the cutoff threshold calculation: is it just based on one negative sample? tested in duplicate?triplicate? we need more information

Tables 1 and 2 are to some degree conveying similar or the same information (like the totals in the first column of Table 1 and the first row of Table 2). I would suggest one table maybe combined? The p-values are far and few between, most values are not statistically significant (either because of the small numbers or minor differences) but they are not stated in the Tables. In fact Table 1 seems to have only p-values for the top row?

There are some typos (in Figure 2 - women - not woman; foreign not foreingn)

Line 36 - maybe predominantly, but not soley food-borne.

Line 232 - maybe predominantly or frequently - not prevalently

Line 252 - plural for Asians and Latin Americans, and as regards to sex.

There is also throughout a question about the use of the word sex - usually refers to the biological state, whereas gender is used to describe the socially constructed roles - it might be biological sex the authors want to refer to?

Line 295 - not sure what to make of the "technical problems" - what does it refer to?

Lines 314 - 318 refer to possible serious limitations of the study and should be more discussed in detail.

Author Response

Dear Reviewer,

We have carefully read all the comments received and would like to thank you for all your suggestions and constructive comments, which allow us to improve our manuscript.

There are some questions that need to answered regarding the methodology;

  • the protocol mentioned newborn being sampled - this would be a very distinct patient group, how were the dealt with in Results (there is no specific mention). Only 2 out of the 7 younger-than-12-month newborn samples gave a result. The result was 1 serotype I-III and 1 serotype II on the sample at birth, so it was the same as testing the mother’s serum. All the others are non-responders.
  • The ELISA protocol is missing a detailed description of the cutoff threshold calculation: is it just based on one negative sample? tested in duplicate?triplicate? we need more information . In agreement with the reviewer, we have added: “We tested in duplicate 31 samples from 31 seronegative/non-immune patients” Line 146.
  • Tables 1 and 2 are to some degree conveying similar or the same information (like the totals in the first column of Table 1 and the first row of Table 2). I would suggest one table maybe combined? The p-values are far and few between, most values are not statistically significant (either because of the small numbers or minor differences) but they are not stated in the Tables. In fact Table 1 seems to have only p-values for the top row? Thank you for the suggestion. However, if the two tables are combined, some information may be lost that allows for a simple interpretation of the data
  • There are some typos (in Figure 2 - women - not woman; foreign not foreingn) the terms have been corrected as suggested
  • Line 36 - maybe predominantly, but not soley food-borne. Delete food-born from the text: “Toxoplasma gondii is a protozoan parasite responsible for human toxoplasmosis (HT) … “ line 36
  • Line 232 - maybe predominantly or frequently - not prevalently. Changed in the text “predominantly”
  • Line 252 - plural for Asians and Latin Americans, and as regards to sex… done in the text .
  • There is also throughout a question about the use of the word sex - usually refers to the biological state, whereas gender is used to describe the socially constructed roles - it might be biological sex the authors want to refer to? We agree with the reviewer. The word sex refers to the biological state. …..we modified this term at line 278,281
  • Line 295 - not sure what to make of the "technical problems" - what does it refer to? As reported in line 290 the effects of multiple refrigeration cycles or poor storage of sera with consequent repeated thawing can affect the sensitivity of the test and therefore reduce the ability of antibodies to recognize strains
  • Lines 314 - 318 refer to possible serious limitations of the study and should be more discussed in detail.  In agreement with the reviewer, we have added: “Fourth, there may have been a selection bias for the females as they were probably asymptomatic while pregnant, and for the males because they were mostly symptomatic. This suspicion cannot be confirmed as we had only clinical judgements, and not clinical data. Lines 337-340

Round 2

Reviewer 2 Report

Authors answered satisfactorily most of my comments. Though, I believe that since new information was added in lines 92-93 “These two institutions are primary referral centers for HT in pregnancy and parasitic diseases in Italy”, text lines 352-354 should be revised accordingly to cover any potential contradiction.

Author Response

Dear reviewer, 

In agreement with your observation, we have changed this sentence at line 352-354: 

"Finally, although the study was conducted in two Italian reference centers for HT in pregnancy and parasitic diseases, it was conducted on a small population, preventing the generalization of the results".

Reviewer 3 Report

Thank you for the revision

Author Response

Thanks for your valuable comments